# Cartographic Design and Processing of Originally Printed Historical Maps for Their Presentation on the Web

**Petra Justová** * and **Jiří Cajthaml**

Department of Geomatics, Faculty of Civil Engineering, Czech Technical University in Prague, Thákurova 7, 166 29 Praha, Czech Republic; jiri.cajthaml@fsv.cvut.cz
* Correspondence: petra.justova@fsv.cvut.cz

**Abstract:** On the example of our project on the creation of the historical web atlas on Czech history, we introduce the process of adapting originally printed historical maps for their presentation in the web environment, which overcomes the shortcomings of standard approaches in similar projects based on printed predecessors published only as zoomable scanned analogues or default GIS maps. To simplify the originally complex map and to increase the information potential of the maps, we propose seven different types of additional map functionality according to the specific characteristics of the original map content. In addition, we present a set of rules, principles, recommendations, and methods for the cartographic design and processing of originally printed historical maps that should be considered when they are prepared for presentation on the web, including the description of the specific visualisation processes for the proposed types of map functionality. The proposed complex methodology can be applied to similar projects focused on the conversion of originally printed maps to the web and may contribute to improving the quality of the visualisation and presentation of historical maps on the web in general.

**Keywords:** web cartography; cartographic design; map processing; historical map; historical atlas; printed maps; web map application; map functionality

## 1. Introduction

### 1.1. Visualisation of Historical Data

Historical events and processes are inherently related to a specific time and geographical space. When visualising historical data in maps, it is essential to deal primarily with cartographic methods for depicting development over time, because capturing a change in the spatial determination of a phenomenon or a change in the spatial relationships between the depicted phenomena over time is crucial when studying (and presenting) history. In the creation of historical maps, the need to capture multiple temporal states of a given phenomenon is very common, so the knowledge of methods suitable for visualising time in maps is an essential skill for the historical cartographer. The problem of visualising time as a fourth dimension in the two-dimensional map space is summarised by Monmonier [1], who compares the limits of time representation in static maps with the possibilities of its representation in dynamic maps, or by Vasiliev [2], who defines several categories of time according to the type of temporal information. His work is based on the study of earlier map works and presents a comprehensive overview of the methods of the cartographic representation of spatio-temporal information depending on the type of temporal information and the spatial dimension of the phenomenon. His research builds on previously published works on similar topics [3,4]. The difficulty of representing time within a static medium is also discussed by Kraak [5], who describes several temporal concept models and presents multiple cartographic solutions to depict the spatio-temporal information in a map. Wigen and Winterer [6] provide a historical overview of cartographic efforts to represent time on maps. The effective representation of time in dynamic maps is also addressed by other authors [7,8].

A specific problem in visualising and presenting historical data is the representation of the location or distribution uncertainty of a phenomenon [9]. This problem is caused by the way data are acquired during historical research (positional identification of an object or delineation of the area of occurrence of a phenomenon based on textual information, extraction of data from positionally inaccurate map bases). The issue of displaying the position or attribute uncertainty in a GIS environment has long been addressed by numerous authors [10,11].

### 1.2. Printed Historical Atlases

Historical atlases represent a very specific and interesting type of historical cartographic work. They reflect not only the theoretical and methodological level of the discipline, or the development of historical discourse but also the applied methods of the cartographic representation of historical events and related phenomena. Via school education historical atlases, these works can penetrate into wider society and form the way history is perceived and evaluated by the public [12]. This fact has been grossly abused by political forces, especially in totalitarian systems [13]. The analysis of the approaches to the cartographic processing of historical phenomena in previously issued atlases with a similar focus can become a great source of inspiration and is strongly recommended when compiling an atlas [14–16].

Historical cartography began to develop rapidly in the 19th century when the modern form of the historical atlas began to take shape and the tradition of the first historical school atlases was established [17,18]. This advance continued also in the 20th century, especially after World War II. During this period, more specialised works appeared, and there was also an increase in the range of topics covered in general historical atlases [19,20].

### 1.3. Electronic Historical Atlases

At the end of the 20th century, the technological progress and the use of digital methods in cartography radically affected the way of the processing and publication of atlases that began to transform from original paper media to digital media. Initially, electronic atlases were released as stand-alone applications, but in the 1990s, due to the progress of the Internet, data transfer standards, and the development of graphic formats, the first electronic historical web-based atlases began to appear [21]. The earliest web-based historical atlases were often developed in parallel with the creation of their analogue version and were presented as browsable applications with limited functionality. Along with the digital revolution, new visualisations and functionalities have entered the field of atlas cartography, and web-based atlases have started to be enriched with a related multimedia content (diagrams, text, visuals, videos, and animations) linked to the displayed geographic entities and have gradually evolved into multimedia atlas information systems (MAIS) [22]. The process of creating web-based atlases and its development over time has been addressed in the studies by Swiss researchers from the Institute of Cartography in Zurich [23–25], along with the discussion on different approaches used in the development of web atlases, which were based on multimedia content, but without including cartographic aspects. The importance of including the cartographic aspect in the process of creating a web-based interactive atlas was also emphasised by Lechthaler [26], who mentions the term cartographic information system (hereafter CIS) in this context. CISs are primarily focused on a high-quality cartographic visualization of the displayed data [27], which includes the additional processing of the presented information, where basic cartographic rules are applied to the displayed data [28].

Over the last three decades, two basic approaches to the creation of historical web atlases have evolved: (1) electronic versions of previously existing printed atlases and (2) stand-alone electronic atlases. They differentiate in the way and the quality of the cartographic visualisation and presentation of data, in the kinds of map interaction/functionality or in the thematic and geographical scope. In general, historical web atlases that were developed in parallel with or as follow-ups to their printed versions present a broad thematic

focus of maps (population, economics, culture, military, etc.). They are focused on national or local history (regions or towns) and provide a more accurate historical content. They are often processed in the form of so-called web map portals presenting either zoomable default GIS maps [29] or georeferenced scanned analogue maps enhanced with simple animation [30] or combining scanned analogue maps or raster images with vector graphics and enabling the user to change the map content via layer selection [31]. The later mentioned map functionality is very common for historical town atlases [32–34]. In contrast, stand-alone electronic historical atlases present topic-specific maps on the territorial and political development of the world or its part over a certain time period. Most of them are distributed in the form of non-zoomable static raster images with temporal animation, map switching, or hyperlinks [35–38], or in the form of animated videos composed of static raster images [39]. Only a few examples of stand-alone electronic atlases follow the specific rules of web cartography, which result from the specific characteristics of the web as a presentation medium [40], and present their content as dynamic multiscale vector maps and offer the user the basic web map functionality (zooming, panning, pop-up display, and hyperlinks) or an interactive timeline [41–43].

Many of the above-mentioned projects were developed more than 15 years ago, which is reflected in the technological aspects of the map applications that directly affect their functionality. Some of them (developed in the late 1990s) are even no longer updated and are technologically obsolete [31,44,45]. The data preparation and cartographic processing in most of the above examples were performed in desktop GIS software using standard data formats for the conversion to the web environment. For the web presentation, the authors mostly used a combination of HTML client programming, CSS, and JavaScript. The dynamics and interactivity of web maps were programmed using the open-source libraries Open Layers or Leaflet, or within commercial APIs (ArcGIS API for JavaScript, MapBox GL JS).

### 1.4. Aims

In this article, we present a set of rules, principles, recommendations, and methods for the cartographic design and processing of historical maps (originally prepared for the print medium) for their presentation in the web environment. The procedure is demonstrated on the example of our recent project on creating the historical web atlas [46], which presents selected historical maps from two printed historical atlases [47,48]. The main aim of our work was to create a historical web atlas that overcomes the shortcomings of similar currently existing historical web atlases (created as electronic versions of previously existing printed atlases) that mostly display the originally printed maps on the web in the form of raster images without adequately adjusting their cartographic design to the specific characteristics of the web as a presentation medium (see Section 2.2).

In this article, we focus on the description of selected cartographic aspects of the visualisation and presentation of historical maps (originally prepared for the print medium) in the web environment. We do not deal with the map interface design and its effectiveness (form, placement, or visual hierarchy of layout elements) [49,50] or with the overall conception of an atlas [51,52] nor with the data, software, or technological aspects of web maps.

The sub-objectives of our project can be summarised as follows:

- To create a historical web atlas that would differ from standard approaches in similar projects that present only digitized analogue maps;
- To design the type of web map functionality (map dynamics and interactivity) according to the map characteristics to increase the information potential of original maps;
- To propose a methodology for the conversion of originally printed historical maps to the web environment;
- To keep the cartographic quality of the original printed maps with respect to the specific characteristics of the web as a presentation medium.

## 2. Methods

The process of adapting the originally printed historical maps for their presentation in the web environment was solved in several subsequent stages:

- Map analysis (Section 2.1);
- Cartographic design and map processing (Section 2.2);
- Map publishing (Section 2.3).

### 2.1. Map Analysis

During the initial stage of the preparation of our historical web atlas, an analysis of the maps was carried out to identify the most suitable type and the level of web map functionality that would simplify the originally complex map while enabling a meaningful interaction with the original map content in the web environment that would increase the information potential of the presented map. In the detailed map analysis (Figure 1), we identified three basic categories of maps with similar characteristics of the data component type and map complexity (visual, intellectual, or both) [53]. According to these characteristics, we assigned the appropriate map functionality to each map. Our approach was inspired by the concept of restrictive flexibility when the users explore the map due to predefined interactive functions to fulfil the intended information task of the presented map and to ensure the legibility of the map in all phases of interaction with its content [54].

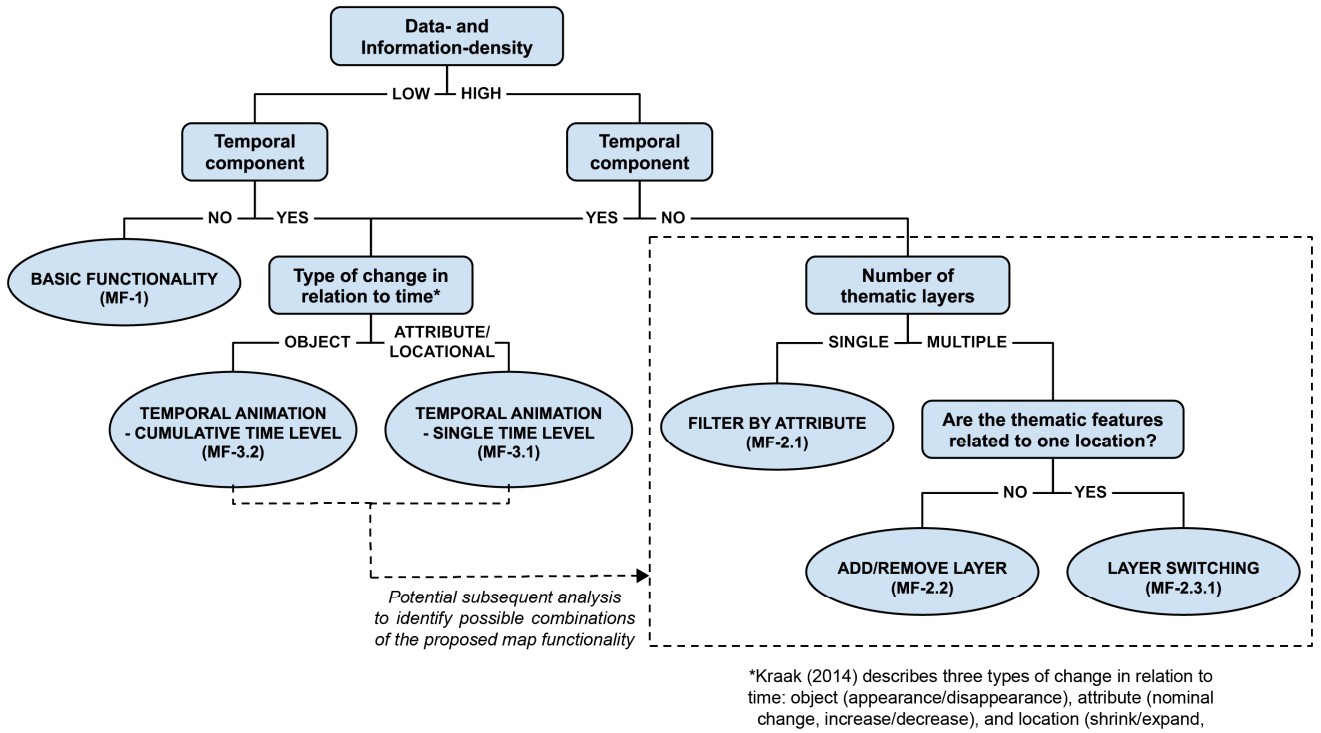

**Figure 1.** Map analysis process [5].

For the category of simple analytical maps with low data- and information-density, no inset maps, and no temporal data component, we decided to apply only the basic map functionality (MF-1): zooming, panning, feature highlight, and pop-up display, as an additional functionality would not add any more informational value to the user. This set of map interactions was defined as the minimum level of functionality that was applied to all web maps as a default. An additional map functionality was applied to more complex maps with a high data- or information-density.

Based on the similar characteristics of maps, we determined the following types of additional map functionalities:

- Filter by attribute (MF-2.1): The user has the option to filter the features according to the predefined attribute to analyse the spatial pattern/distribution of phenomena and their relations. This functionality is suitable for high data-density maps displaying a single thematic layer of qualitative data with no temporal component.
- Add/Remove layer (MF-2.2): The user is allowed to customise the map content by adding or removing thematic layers while preserving the initial default thematic layer in the map view. This functionality can be applied to maps displaying multiple thematic features with no temporal component that are not related to one location or are of different spatial dimensions.
- Layer/Map switching (MF-2.3): The user can switch the displayed thematic layer (MF-2.3.1) or change the geographical extent of the map (MF-2.3.2). The first mentioned option is suitable for maps displaying multiple thematic layers/features with no temporal component related to one location that are represented by multiple symbols of the same spatial dimension, or for multivariate maps combining different thematic methods. This layer switch option enables the user to reveal the thematic information in a separate visually simplified map while keeping the thematic information of the remaining (not visually represented) layers in a pop-up window. Map switching functionality (MF-2.3.2) is best useable to show more detail of a portion of the main map (originally displayed in an inset map) in a single map view.
- Interactive temporal animation (MF-3): The user can control the position of the slider on the timeline with predefined time sections. Temporal animation enables the user to better perceive the development of depicted phenomena over time as it is presented either as a sequence of maps representing a single time level (MF-3.1) or as a sequence of maps presenting individual time levels cumulatively (MF-3.2). The former option can be applied on maps displaying several states of phenomena in multiple time levels, as the state of the depicted phenomena is valid only for the displayed time interval (e.g., administrative border or military campaign). On the contrary, the cumulative sequence is suitable for maps displaying features with temporal information on their appearance and disappearance, as many features existed over several time intervals (e.g., development of a railway network).

Temporal animation was also applied on a set of simple analytical maps with no temporal data component that capture the state of a phenomenon at one moment in time, but together form a temporal sequence (e.g., political maps).

### 2.2. Map Processing

The cartographic design and map processing for presentation in a print medium differ greatly from those required for the presentation of maps in the web environment. These differences are mainly due to different characteristics of the web as a presentation medium (hypermedia, interactivity, dynamics, and unlimited map extent) and due to different requirements of the display device (RGB colour model and limited resolution) [55]. Therefore, when maps that were originally prepared for the print medium are to be presented on the web, they must be adequately adjusted to the specific characteristics of this environment, which often requires considerable intervention in the design of the original maps, which were not initially prepared for this purpose.

To meet the requirements for the presentation of maps on the web, the map processing procedure was divided into five consecutive stages that correspond to specific characteristics of the web environment:

- RGB colour mode and symbol simplification (Section 2.2.1);
- Zooming (Section 2.2.2);
- Panning (Section 2.2.3);
- Pop-up window (Section 2.2.4);
- Additional map functionality (Section 2.2.5).

### 2.2.1. RGB Colour Mode and Symbol Simplification

At the beginning of map processing, we modified the original map key in terms of the colour mode and the complexity of symbols. To preserve the colours of the printed atlas for the web environment, the original CMYK colour model was converted to its RGB equivalent. The complex cartographic symbology was simplified as it could be replaced by the functionality of the web map (Figure 2). The qualitative or quantitative characteristics of the feature were displayed only in the pop-up window. The temporal information about the year of the appearance or disappearance of a feature was displayed in the timeline.

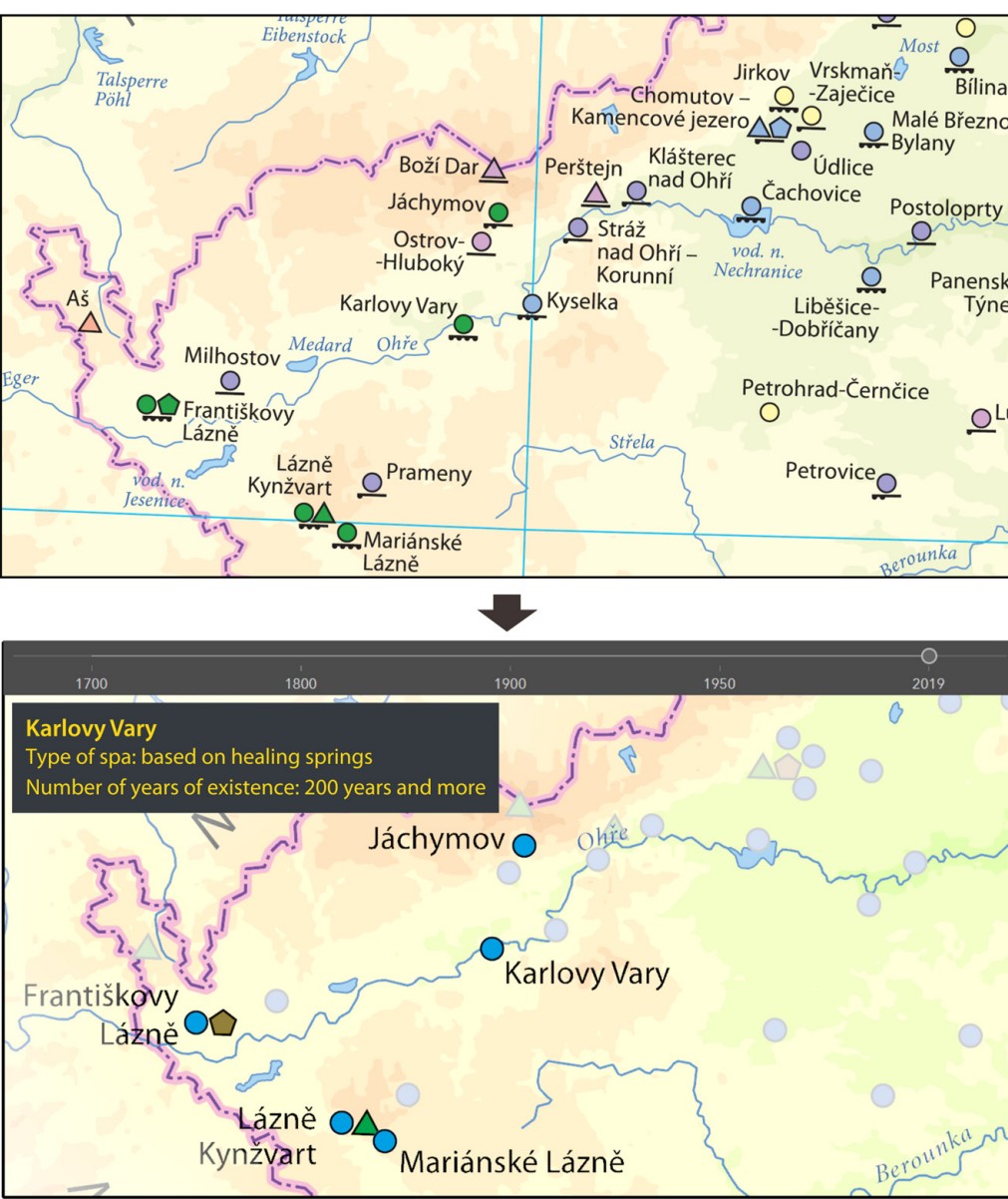

**Figure 2.** Complex cartographic symbology of the original map (**top**) vs. simplified map symbology replaced by the web map functionality (**bottom**).

### 2.2.2. Zooming

In the next stage, it was necessary to redesign the initially static maps that were designed at a fixed scale into dynamic multiscale maps. At first, we determined the basic scale to optimally display the whole area of interest in the respective historical time on a common computer screen. Considering the different extents of the territory of the Czech Lands in different historical periods, two basic scales had to be chosen (1:3M for

Czechoslovakia and 1:2M for the Czech Republic or the Czech Lands). This fact affected the way in which other zoom levels were derived (ratio of 1.5) as we aim to unify the scale series to provide a visual comparison of the web maps within a single web map application. For the same reason, we also unified the map projection, since the original maps used the Albers equal area conic projection but with a different position of the central meridian depending on the geographical extent of the historical territory of the Czech Lands. The maximum zoom level was set specifically for each web map with regard to the level of detail and generalisation of the original data. For each zoom level, we set the symbol and label size with respect to the technical limitations of display devices to improve the legibility of the web map.

After setting the symbol and label size for predefined zoom levels, we proceeded to edit the original map projects for a dynamic multiscale display on the web using the modified original map key (RGB colour model, simplified symbols, and symbol and font size for each zoom level). It was necessary to redesign the original map composition by symbol and label replacement to avoid the symbol overlap or the ambiguity of label association that appeared frequently due to the modification (enlargement or reduction) of the original size of cartographic symbols and labels. Automatic dynamic label placement was not applied, as the result did not meet our requirements for a high-quality cartographic design and information value of the maps [56,57]. Along with the symbol and label replacement, we reduced the visual complexity and information density of some maps at smaller scales: (1) by removing the labels showing the name or time attribute of a thematic feature, as these characteristics could be displayed in a pop-up window, (2) by the selection of toponyms according to their thematic importance, or (3) by aggregating the symbols. The latter step was discussed with the subject specialist, historian, or historical geographer. The original labels have been preserved only at the largest viewable scale.

### 2.2.3. Panning

In addition to processing the maps for zooming (see above), it was necessary to adequately process the maps for the application of the remaining basic map functionality, panning, and pop-up display. As the original maps were designed for presentation in a print medium, the map content was restricted only to the predefined map frame. Although web technologies offer different methods to limit the extent of the web map that the user can explore, none of them completely solved our problem of missing data at the edge of the original map frame. Therefore, we decided to apply a covering polygon layer outside the area of interest (thematic map content) using the feathering effect to cover the area with no data.

### 2.2.4. Pop-Up Window

The attribute data in the geodatabase of the original maps were often incomplete or in an inappropriate format (abbreviations or numerical codes) as they were primarily used for visualisation. Therefore, it was necessary to edit or complete selected attributes that were intended to be displayed in a pop-up window.

### 2.2.5. Additional Map Functionality

Besides the above-mentioned general modifications in the cartographic design of maps, it was also necessary to redesign the original visualisation and data structure of some maps according to the type of additional map functionality:

- Filter by attribute (MF-2.1): A copy of the filtered thematic layer with faded symbology was made to visually distinguish inactive features while filtering. This duplicate layer also preserved the original thematic content in the background in all phases of interaction with the map.
- Add/Remove layer (MF-2.2): No additional visualisation process was applied. For the purpose of this functionality, we only differentiated the thematic layers by assigning a unique ID that specified the initial setting of the layer visibility.

- Layer/Map switching (MF-2.3): Originally polythematic or multivariate maps had to be decomposed into several simplified monothematic maps that could be displayed separately in a single map frame (MF-2.3.1). This process often required a complete redesign of the original map composition either by symbol and label replacement (Figure 3), or by modifying the original thematic cartographic method. Inset maps that were going to be displayed in a single map frame (MF-2.3.2) required a more radical intervention in the original map composition as the final scale significantly differs from the original one.

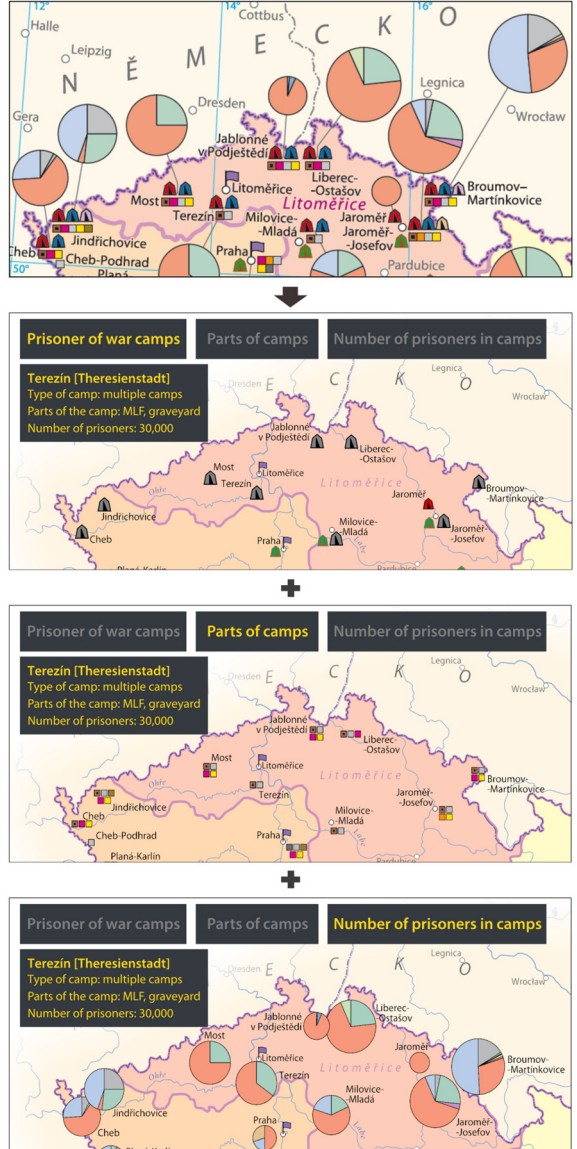

**Figure 3.** Decomposition of the original polythematic and multivariate map with the detail of the pop-up window showing the thematic information of all original layers.

- Interactive temporal animation—single time level (MF-3.1): Single maps that displayed several states of phenomena in multiple time levels were decomposed into a series of maps, each displaying the state of the phenomena only in a given time interval. If the temporal component of the data was expressed by means of visual variables (e.g., change of administrative borders), it was necessary to redesign the original cartographic visualisation and create new visualisations for each time interval (Figure 4). If the temporal component of the feature was expressed only through the exact time annotation (movement of the army or a battle on military maps), the

visualisation for each map view was created by filtering the features according to the time or another predefined attribute. The predefined attribute represented the order of the time interval on the timeline and was derived from the time attribute in the geodatabase. Very often it was necessary to complete the original database of the map, as temporal information for some features was missing. The time intervals (sections of the timeline), their number, and range were set according to important milestones in the development of the depicted phenomena. This process required a discussion with the subject specialist, historian, or historical geographer. In some cases, this information could be easily retrieved from the original map (exact time determination of the change of the borders or the movement of the war front). In the temporal animation of military maps, we decided to visually preserve the previous state of phenomena in the map view to facilitate the comprehension of the spatial context of the movement. For this purpose, the duplicate of the thematic layer with faded symbology was created, to visually distinguish the current and the past time interval (Figure 5).

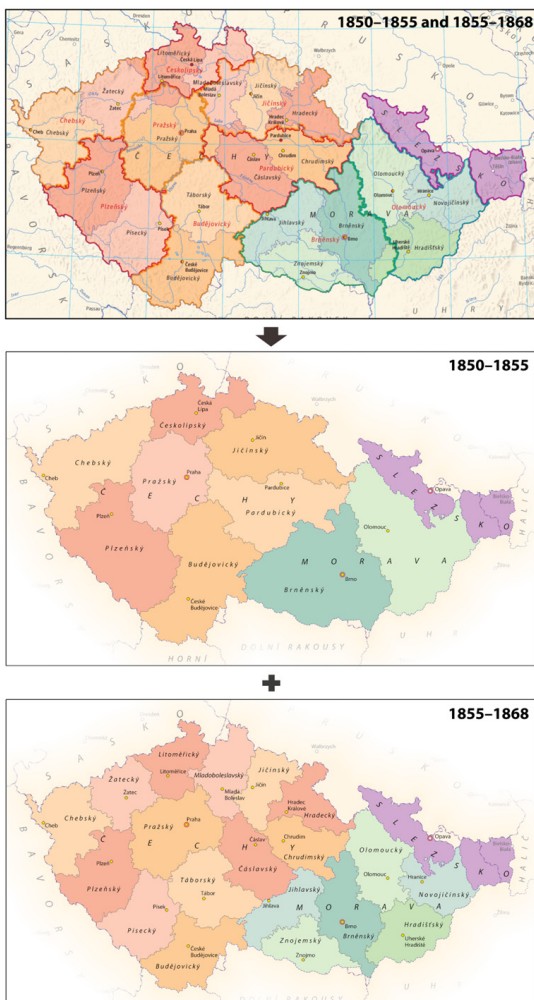

**Figure 4.** Decomposition of the original map displaying several states of phenomena in multiple time levels by means of visual variables.

- Interactive temporal animation—cumulative time levels (MF-3.2): The complex cartographic symbology of maps that originally displayed features with temporal information on their appearance and disappearance (often along with many other attributes) had already been simplified within the general modifications of the cartographic design (see above). Therefore, the original map composition could be decomposed into

multiple temporal views only by filtering features according to the time or another predefined attribute (see above). In contrast to the afore-mentioned functionality (MF-3.1), each temporal view included the content of the previous one, as many displayed features existed over several time intervals (Figure 6). The faded symbology was applied only for disappeared features.

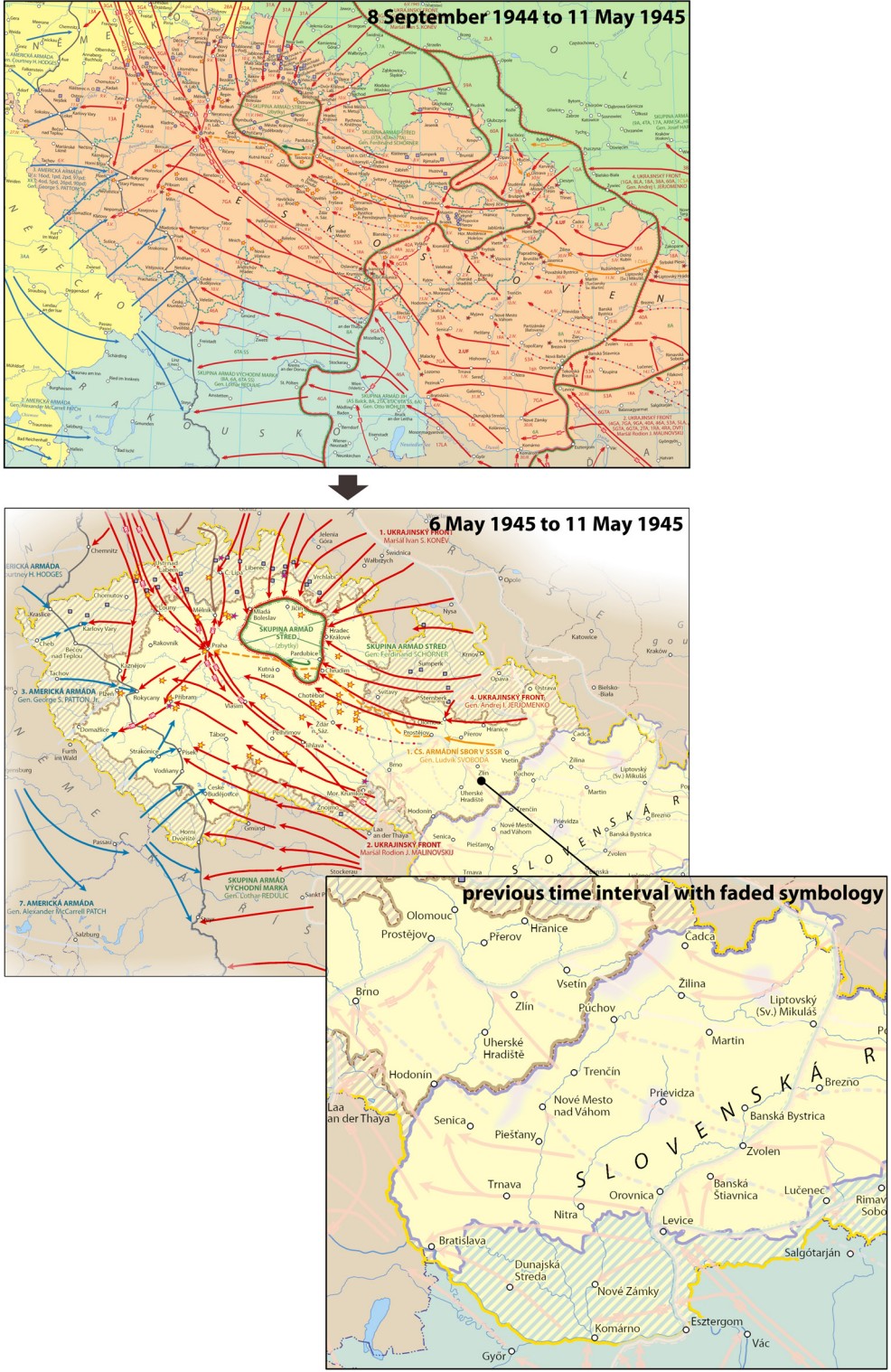

**Figure 5.** Faded symbology used in a military map to visually preserve the previous state of phenomena in the map view to facilitate the comprehension of the spatial context of the movement.

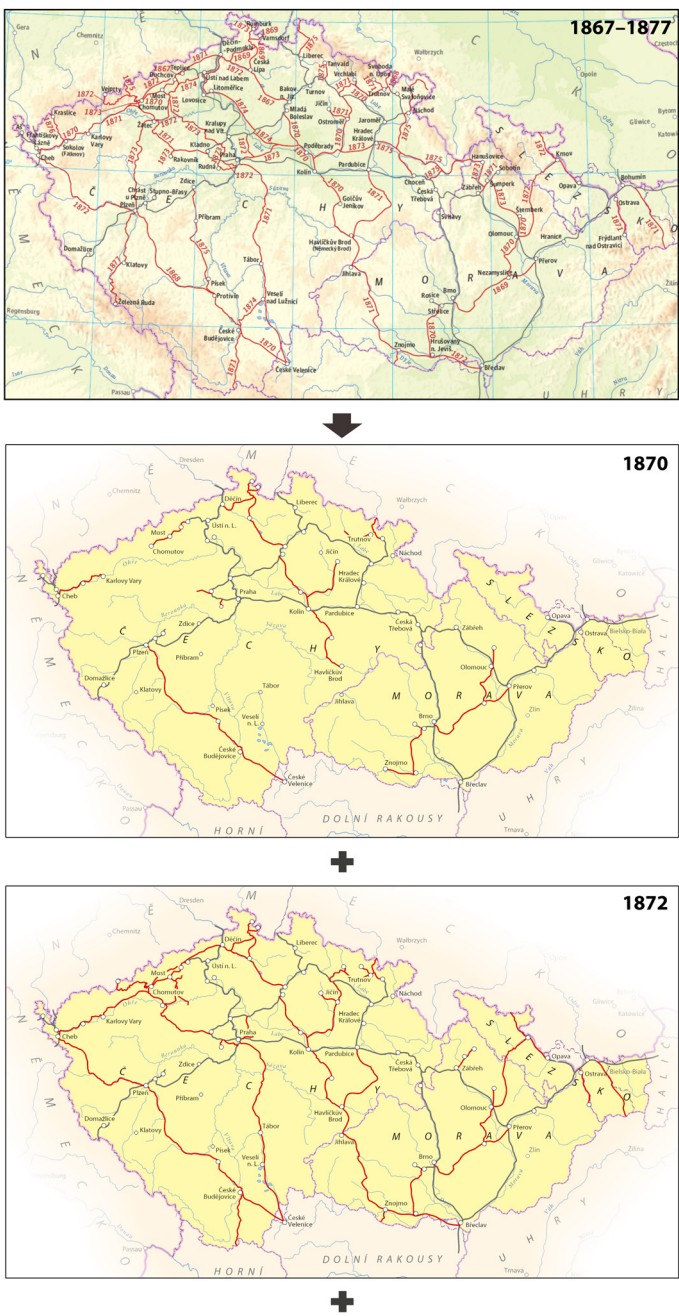

**Figure 6.** Decomposition of the original map displaying features with temporal information on their appearance into multiple temporal views that present individual time levels cumulatively.

### 2.3. Map Publication

To publish maps on the web, the map layers were divided into two groups (Figure 7):

- Thematic layers—an active content of the map over which a certain functionality is applied;
- Background layers—an inactive content of the map.

All map layers have been assigned a unique ID that specified the layer behaviour in the web map, such as which layers are jointly controlled by the elements of the additional map functionality (e.g., display in the same section of the timeline or joint switching on/off) or the initial setting of the layer visibility (MF-2.2).

Due to the limitations of the technology used for publishing the maps, a compromise solution in the cartographic design had to be adopted in some cases, as certain layers

or drawing techniques could not be published in the web environment properly (e.g., annotation groups or masks).

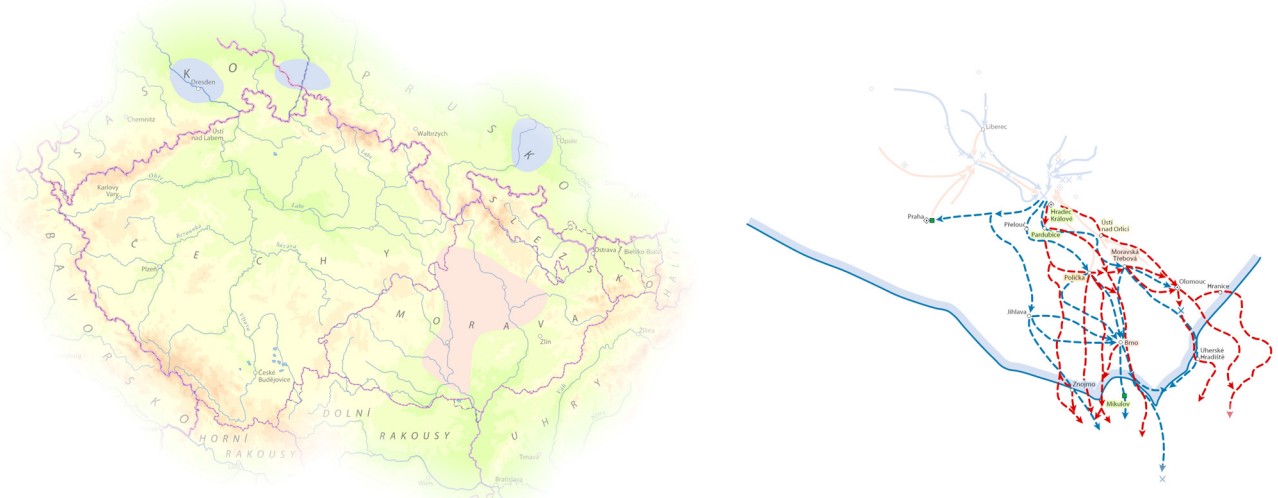

**Figure 7.** Background (inactive) layers of the map (**left**) vs. thematic (active) layers over which a certain functionality is applied (**right**).

## 3. Results

On the example of our recent project on creating the historical web atlas on Czech history [46], we introduced the process of adapting originally printed historical maps for their presentation in the web environment. During the initial stage, we carried out a detailed analysis of more than 160 maps to design the appropriate type of map functionality for each map that would simplify the original complex map (both visually and intellectually) and increase the information potential of the presented map (in comparison to its printed version). Based on the analysis (Figure 1), we identified seven categories of maps with similar characteristics of the data component type and map complexity (Figure 8). According to these map characteristics, we designed a web map application template with a predefined type of map functionality for each map category. The types of map functionalities were determined as follows:

- Basic functionality (MF-1);
- Filter by attribute (MF-2.1);
- Add/Remove layer (MF-2.2);
- Layer switching (MF-2.3.1);
- Map switching (MF-2.3.2);
- Interactive temporal animation—single time level (MF-3.1);
- Interactive temporal animation—cumulative time levels (MF-3.2).

In the next stage, we focused on the cartographic design and processing of the maps for their publication on the web. At first, we have described the general modification of maps that should be considered when preparing the map for its presentation on the web and that corresponds with the basic characteristics of the web environment (interactivity, dynamics, and unlimited map extent) and technical requirements of the display device (RGB colour model and limited resolution).

The general steps of map processing with basic recommendations can be summarised as follows:

- Conversion of the original CMYK colour model to its RGB equivalent;
- Simplification of complex cartographic symbology (replacement by map functionality);
- Redesign of the originally static map to a dynamic multiscale map: (1) setting the maximum zoom level according to the level of detail of the original data; (2) reducing the visual complexity and information density of some maps at smaller scales;

— Solving the problem with missing data at the edge of the original map frame: (1) applying a method to limit the extent of the web map; (2) applying a covering polygon layer;
— Editing or enrichment of attribute data of the original maps for displaying additional information in pop-up.

| map functionality | component type | map characteristics | description | additional visualization process |
|---|---|---|---|---|
| **Basic functionality (MF-1)** | locational attribute | ▪ simple cartographic symbology ▪ low data-density maps | ▪ zooming ▪ panning ▪ pop-up display | none |
| **Filter by attribute (MF-2.1)** | locational attribute | ▪ qualitative data (two or more thematic categories) ▪ high data-density maps | ▪ filtering the features according to the predefined attribute ▪ user can analyse the spatial distribution of phenomena and their relations | ▪ duplicate layer with faded symbology for inactive features while filtering |
| **Add/Remove layer (MF-2.2)** | locational attribute | ▪ multiple thematic features not related to one location ▪ multiple thematic features of different spatial dimension | ▪ customisation of the map content by adding or removing thematic layers ▪ preserving the initial default thematic layer in the map view | none |
| **Layer switching (MF-2.3.1)** | locational attribute | ▪ multivariate map combining different thematic methods ▪ multiple thematic layers/features related to one location represented by multiple symbols of the same spatial dimension | ▪ switching the displayed thematic layer ▪ user can reveal the thematic information in a separate visually simplified map while keeping the thematic information of the remaining (not visually represented) layers in pop-up | ▪ decomposition of the original polythematic (multivariate) map into single monothematic maps ▪ redesign of the original map composition: (1) by symbol and label replacement, or (2) by modifying the original thematic cartographic method |
| **Map switching (MF-2.3.2)** | locational attribute | ▪ two or more inset maps | ▪ change the geographical extent of the map ▪ user can show original inset maps in a single map view | ▪ may require more radical intervention in original map composition (symbol and label replacement) as the final scale significantly differs from the original one |
| **Interactive temporal animation––single time level (MF-3.1a)** | locational attribute temporal | ▪ maps displaying several states of phenomena in multiple time levels ▪ temporal component expressed by means of visual variables | ▪ timeline with predefined time sections ▪ the development of depicted phenomena over time is presented as a sequence of a maps representing single time level | ▪ decomposition of the original map into multiple temporal views by modifying the original thematic cartographic method |
| **Interactive temporal animation––single time level (MF-3.1b)** | locational attribute temporal | ▪ maps displaying several states of phenomena in multiple time levels ▪ temporal component expressed via exact time annotation | ▪ timeline with predefined time sections ▪ the development of depicted phenomena over time is presented as a sequence of a maps representing single time level | ▪ decomposition of the original map into multiple temporal views by filtering features according to the time or other predefined attribute ▪ duplicate layer with faded symbology for past time interval |
| **Interactive temporal animation––cumulative time levels (MF-3.2)** | locational attribute temporal | ▪ maps displaying features with the temporal information on their appearance and disappearance | ▪ timeline with predefined time sections ▪ the development of depicted phenomena over time is presented as a sequence of maps presenting the individual time levels cumulatively | ▪ decomposition of the original map into multiple temporal views by filtering features according to the time or other predefined attribute ▪ duplicate layer with faded symbology for disappeared features |

**Figure 8.** Typology of map functionality according to the specific characteristics of the original map content, including the description of the specific visualisation process.

Besides the general modifications, we have also outlined specific visualization processes for each type of map functionality (Figure 8). Additional visualization processes were suggested as follows:

- Using faded symbology to visually distinguish active and inactive features while filtering (MF-2.1) for existing and disappeared features at a given time interval (MF-3.2);
- Redesign of the original map composition either by symbol and label replacement (MF-2.3.1) or by the modification of the original thematic cartographic method (MF-2.3.1, MF-3.1a);
- Creating a visualisation for temporal views by filtering features according to the time or another predefined attribute (MF-3.1b, MF-3.2).

## 4. Discussion and Conclusions

In comparison with national web-based atlases [24,28], which are created mainly as stand-alone web products based on originally unvisualised GIS data, historical web atlases are very often created as follow-ups to their printed version. In addition, historical atlases present more visually and intellectually complex maps that require a specific approach in terms of their cartographic design and map processing for their presentation on the web (compared to simple thematic maps in national atlases).

Currently, there exists no comprehensive study or research that addresses the issue of the cartographic design and processing of historical maps (originally prepared for the print medium) for their presentation in the web environment. Within our recent project on the creation of the historical web atlas on Czech history [46], we aimed to create an interactive version of the historical web atlas that would differ from standard approaches in similar projects based on printed predecessors that publish scanned analogues or default GIS maps in the form of zoomable raster images [29–31] and that would offer the user an additional map functionality that would increase the information potential of the presented maps. All of the abovementioned examples use the simplest way of displaying maps on the web and do not respect the different characteristics of the web as a presentation medium that require different approach to cartographic design and map processing. If the additional map functionality is applied, it is applied universally to all maps and does not respect the specific characteristics of presented maps.

Therefore, we decided to design the appropriate type of map functionality according to the specific characteristics of the original map content to increase the information potential of the maps. Based on the defined categories of maps with similar characteristics of the data component type and map complexity, we proposed seven different types of additional map functionality suitable for the presentation of historical maps on the web. In addition, we presented a set of rules, principles, recommendations, and methods for the cartographic design and processing of originally printed historical maps that should be considered when preparing the map for presentation on the web, including the description of the specific visualisation processes for the proposed types of map functionality.

Although the map functionality typology and the proposed methodology were derived on the basis of a sample of more than 160 historical maps on Czech history, considering the similar thematic structure of historical atlases [58], we assume that this sample is sufficiently broad and representative that our proposed principles, methods, and recommendations can be applied to other similar projects focused on the conversion of originally printed (digitally created) maps to the web.

In our project, a single map functionality was assigned to each map. We are aware that the application of single map functionality may not be sufficient for more complex maps with temporal data component. Therefore, we suggest a further analysis of maps with temporal data components in terms of the number of thematic layers and the characteristics of the thematic features (Figure 1), that was originally performed only on maps with no temporal component. Based on this further analysis, possible combinations of the proposed types of map functionality can be identified.

By publishing our experience in creating the historical web atlas, we aim to encourage authors from the field of historical cartography to publish originally printed maps/atlases on the web to make them available to a wider audience and to take advantage of the web map functionality for presenting complex historical topics by means of tools that significantly increase the information potential of the displayed content, which a printed or scanned analogue map (or a default GIS map) is unable to provide. Nonetheless, for some projects based on printed predecessors that exist only in analogue form, this may require a very time-consuming vectorisation process of the original maps.

The proposed map functionality typology and the complex methodology on the cartographic design and processing of originally printed historical maps for publication on the web can contribute to the improvement of the quality of the visualisation and presentation of historical maps in the web environment. Some general principles may be applicable even for the creation of stand-alone (historical or non-historical) web projects and provide a guidance for the presentation of complex maps on the web.

In further research, the proposed map functionality and the additional visualisations applied are planned to be evaluated in terms of information transfer efficiency to the target user. For this purpose, a series of usability tests will be performed using the knowledge testing or eye-tracking methods [59,60] to better understand the users of the atlas by researching who they actually are, how they interacted with maps, and for what purpose they used the maps [61,62].

**Author Contributions:** Conceptualisation, Petra Justová and Jiří Cajthaml; methodology, Petra Justová; writing—original draft preparation, Petra Justová and Jiří Cajthaml; writing—review and editing, Petra Justová and Jiří Cajthaml; visualisation, Petra Justová; supervision, Jiří Cajthaml; project administration, Jiří Cajthaml. All authors have read and agreed to the published version of the manuscript.

**Funding:** This work was supported by the Grant Agency of the Czech Technical University in Prague, Grant SGS23/051/OHK1/1T/11 "Analysis, visualization and presentation of 2D and 3D geospatial data using modern methods of cartography and GIS".

**Data Availability Statement:** Not applicable.

**Conflicts of Interest:** The authors declare no conflict of interest.

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
