# Peer review of "Cartographic Design and Processing of Originally Printed Historical Maps for Their Presentation on the Web"

_ijgi, doi:10.3390/ijgi12060230_

Round 1

Reviewer 1 Report

Very good article, actual and modern research in cartography, well structures, clear methods explanations, useful results.

May be will be good if the graticules will be represented on thematic layers of the maps for easier localization and orientation.

Author Response

Thank you very much for taking the time to assess our manuscript. All the comments have been addressed in the attached document.

Reviewer 2 Report

The article is methodologically weak. The authors argue that their proposed approach ‘overcomes the shortcomings’ of other projects that actually have these shortcomings (i.e. transfer historical maps into a digital environment without specifically adapting them). Thus, an already obvious advantage is demonstrated. We note that there are very well-developed interactive historical mapping projects. Some examples of such projects are listed in the references, there are more projects alike, e.g., https://www.oldmapsonline.org/ , http://geacron.com/home-en/ . It would be logical to compare the functionality presented by the authors with that of portals of this type and to show original or more effective solutions.

The literature review section is merged with the Introduction, which does not follow the usual structure of a scientific article.  The existing review of previous research, specifically the “Electronic Historical Atlases” section, is too simplistic,  lacking comparative analysis and deeper insights.

In the overview of previous research, the authors mention a need of cartographic methods to represent uncertainty – but do not demonstrate any of such methods in the paper.

In the Section 1.4, the authors describe what they haven’t dealt with, but do not specify what, in their opinion, efficiency and/or modernity means (the authors claim that their proposed solution for transferring historical maps to an interactive environment is effective, but do not support this claim with evidence). Thus, the objectives do not reflect research problem, and in case of #2 “to design the type of web map functionality (map dynamics and interactivity) according to the map characteristics to enable an effective interaction with the map“ – contradict the other statement: „We do not deal with a map interface design and its effectiveness“. The concept “Specific rules of web cartography” is not clarified, therefore, we cannot ascertain whether these rules have been followed.  The following descriptions of solutions in Section 2 are of various methods and solutions that are commonly used in practical web cartography. if any research has added value to these methods or their application, it is not described. The change of symbology in Figure 1 is not methodologically justified – it is not possible to judge about correctness. It has not been demonstrated that decomposing the maps in such a way that they are truly perceived without losing the original information.The example in Figure 3 raises the question of whether the changes are really perceived well when viewing the decomposed images in this way. 

The provided link to the results online https://www.czechhistoricalatlas.cz/ (#44 in reference list) was not accessible – repeating connection timeout adds no credibility.

To sum up, this is a decent description of the practical project, but it lacks a lot to qualify as a scientific article.

Author Response

(The authors gave the same response as above.)

Reviewer 3 Report

As an important resource to study history, historical map is of great significance to its network display. The author based on the project on the creation of the historical web atlas on Czech history, introduce the process of adapting originally printed historical maps for their presentation in the web environment. To increase the information potential of the maps, they propose seven different types of additional map functionality according to the specific characteristics of the original map content. In addition, they present a set of rules, principles, recommendations, and methods for cartographic design and processing of originally printed historical maps.

 As a whole, the author has done a lot of work but the paper focuses much more on the engineering problems, it is much more like a technical report but not a scientific paper. I suggest that the author should analyze their work and pay much more attention on the special characteristics of historic map. Further refine the scientific issues in the work will make this work more instructive to others.

 Some specific comments:

 Comment 1.

The architecture of the introduction needs to be adjusted. Instead of explaining the background, goal of the research, introduce the state of the art of the study is not suitable for the reader to understand.

 Comment 2.

   The paper is mainly to introduce a methodology for the process of adapting originally printed historical maps for their presentation in the web environment. It is better to use a framework chart to show the process clearly.

 Comment 3.

Although the author put forward the methodology in the part of  2.Methods, but it is lack of analysis for the universality in the discussion part if this method can be used for which kinds of historic map. Now the part of discussion is too much repetition from the method part.

Comment 4.

The structure of the whole article needs to be adjusted, and the method part is too much, the author didn’t show how effective is this method on the Web presentation in the result part.

Some specific comments:

1. Figure 1, there isn’t any annotation in the bottom map, is it a good design?

2. Some spelling mistakes in the paper, such as Page 14, line 459, to improving.   

Author Response

(The authors gave the same response as above.)

Reviewer 4 Report

Thank you for the opportunity to read this manuscript and to learn about the authors' new approach to creating interactive historical atlases.

The paper is well written and in good English expression though with a few writing or typographic errors. It is easy to follow and logically ordered. The introduction provided a useful overview of the area, though there were some references that I was surprised were omitted (see my comments below). The manuscript is mostly well illustrated, though I thought a few additional illustrations might further improve the paper. 

Points for improvement in a revision:

1. I was very surprised that several monographs related to historical mapping, and some even specifically about mapping time were not included in the references.

Kraak, M. J. (2014). Mapping time: Illustrated by Minard's map of Napoleon's Russian Campaign of 1812. Esri.

Wigen, K., & Winterer, C. (Eds.). (2020). Time in maps: from the Age of Discovery to our digital era. University of Chicago Press.

2. The manuscript takes a very Eurocentric view of atlases. Also, some argue that Ortelius's Parergon is not really the beginning of historical atlases: 

Goffart, W. (2007). When Did Historical Atlases Really Originate?. Humanities Research Group Working Papers9.

3. There seems to be a stray 'thematic layer 02' label in Figure 2 (second panel).

4. It would be useful to provide an illustration of Layer/Map switching and how the thematic information appears in a popup. (MF-2.3). I'm having a bit of a failure of imagination to think of what this might look like.

5. I wonder why a popup was chosen rather than a mouse-over. The latter can be done with less effort on the user's side.

6. Figure 4: I fine it pretty hard to see the faded symbols in this map, especially because the base map has changed.  

7. Something that would add more value to the manuscript would be to present some the findings of your analysis of the maps in the Czech atlas. Of those 160 maps, how many were assigned to the different levels in your typology? What were the common patterns of co-ocurrence of map functionality?

8. The discussion of uncertainty in historical maps seems a bit peripheral to the rest of the paper -- there is only one example given of how one could indicate uncertainty. I would either elaborate this or omit it.

9. It's not clear why some sources are greatly abbreviated in the reference list while others are not abbreviated at all.

Author Response

Thank you for taking the time to assess our manuscript. The manuscript was checked and edited by a professional English language editor, the writing and typographic errors should be corrected in the revised version. Thanks for pointing out the missing references, we have added both of them (see Response 1 a 2 in the attachment). All the comments have been addressed in the attached document. 

Round 2

Reviewer 2 Report

Thank you for considering some of the comments. The presentation has been improved. Unfortunately, the most explanations given do not convince me that the article is scientific. It could be published as a sound technical report. 

Author Response

Thank you very much for taking the time to assess our manuscript in the second round of revisions. We know that the manuscript is more practically oriented and lacks some aspects of a standard scientific article. Yet we believe (based on our experience in the field) that such a manuscript will contribute to the field of historical cartography and will help to increase the cartographic quality of similar products.

Reviewer 3 Report

The author has revised to address the issues raised. 

Author Response

Thank you very much for taking the time to assess our manuscript in the second round of revisions. 

Reviewer 4 Report

Figure 1 is a useful addition to the manuscript. However, I’m not sure I understand what the empty box with the dashed line represents. Is this meant to indicate that the other dashed box should also be repeated here? If so, some clear labelling of the empty box to indicate this would be useful. Also, if that is the correct interpretation of this figure, then additionally including the three ovals at the bottom of the left is a bit confusing as they are also in the dashed box at the right.

I still find it problematic that the authors state as a matter of fact that historical atlases emerged in Europe at a particular time when this is the view of a single author (Black) and not established fact. Several other authors quoted in the Goffart piece whose citation I provided also argue against the view that Ortelius’s atlas even constituted an historical atlas. Maps and atlases have been used by those in power in many societies, not only in Europe, to depict a recounting of events that advances the in-power group’s agenda. I do not think that things should be published in the scientific literature as being facts when they are not so established in the literature. Goffart is also an historian writing on historical atlases and he comes to different conclusions than does Black. Ultimately, I don’t think it’s even important for your article when the first historical atlas originated. What is important is describing what an historical atlas contains (an account of historical events) and to what purposes such atlases are put (teaching history, advancing acceptance of a given account of history).

Thank you for explaining that the buttons at the top of the new Figure 3 are how the layer switching is implemented. My prior comments also asked about how the popup windows appear. This does not appear to be depicted in any of the figures.

Figure 5 (previous figure 4): If the fading is difficult to see because of the image size, why do you not then reproduce the image at a larger size? I don’t see the point of including a figure that does not show clearly what you are trying to demonstrate with the figure! You could alternatively show a smaller portion of the map, but enlarged so that the fading is clearly visible.

Author Response

Thank you very much for taking the time to assess our manuscript in the second round of revisions. All the comments have been addressed in the attached document. 
